# Supermode Characteristics of Nested Multiple Hollow-Core Anti-Resonant Fibers

Zequan Li [1,2,3], Jiantao Liu [1,2,3], Changming Xia [1,2,3], Zhiyun Hou [1,2,3] and Guiyao Zhou [1,2,3,*]

1   Guangdong Provincial Key Laboratory of Nanophotonic Functional Materials and Devices,
    South China Normal University, Guangzhou 510006, China
2   Guangzhou Key Laboratory for Special Fiber Photonic Devices and Applications,
    South China Normal University, Guangzhou 510006, China
3   SCNU Qingyuan Institute of Science and Technology Innovation Co., Ltd., Qingyuan 511517, China
*   Correspondence: gyzhou@scnu.edu.cn

**Abstract:** Mode-division multiplexing (MDM) can achieve ultra-high data capacity in optical fiber communication. Several impressive works on multicore fiber (MCF), multi-mode fiber, and few-mode multicore fiber have made significant achievements in MDM. However, none of the previous works can simultaneously maintain the transmission loss, chromatic dispersion (CD), and differential group delay (DGD) at a relatively low level. A nested multiple hollow-core anti-resonant fiber (NMH-ARF) has significant potential for applications in MDM. This study proposes a novel NMH-ARF with its structural design based on the traditional single-core nested anti-resonant fiber. We increased the number of nodes between capillaries. By changing the position of the nested tubes, several interconnected areas form when a single core is separated. We investigated the mode-coupling theory and transmission characteristics of this fiber. This fiber structure showed a low sensitivity to bending and achieved a super-low DGD and a super-low confinement loss (CL) at a wavelength of 1.55 μm while keeping CD relatively low.

**Keywords:** mode-division multiplexing; multiple hollow-core anti-resonant fiber; confinement loss; differential group delay

---

## 1. Introduction

High-speed data transmission has attracted interest in optical fiber research in the present big data era. Although the development of time-division multiplexing, wavelength-division multiplexing, polarization-division multiplexing, and space-division multiplexing (SDM) has exponentially increased the data transmission rate in the past decades, meeting the demand for high data capacity is challenging. Meanwhile, the power loss in fiber increases when the data capacity is expended. The traditional single-mode fiber (SMF) has gradually approached the limit of its transmission capacity due to limitations such as high loss, high dispersion, high nonlinearity, and high damage threshold. In recent years, as a branch of SDM, MDM has become increasingly popular as several magnificent works on multicore fiber, multi-mode fiber, and few-mode multicore fiber have made significant improvements to the issues mentioned above. These fibers use the independence of different cores arranged in parallel or the orthogonality between different modes to realize independent multi-input-multi-output (MIMO) channels. The MIMO technology can demultiplex signals from various channels at the receiver. However, several issues remain unsolved. For example, methods exist to reduce loss during long-reach fiber transmission to a super low level, improve the bandwidth to satisfy high-capacity data transmission, and reduce differential group delay, which is the key to reducing the complexity of MIMO digital signal processing (DSP) algorithms and lowering the enormous cost of mass data transmission. In 2019, a multicore fiber achieved an attenuation of 0.170 dB/km at 1550 nm, which is the lowest loss observed in existing studies on solid-core fibers [1]. Furthermore,

in 2015, a multicore fiber kept the differential group delay below 3.2 ps/km in the whole C-band [2]. In 2016, a multicore fiber kept the differential group delay below 0.5 ps/km at 1550 nm [3]. These records have not made significant breakthroughs before the NMH-ARF proposed in this study. We selected anti-resonant fibers (ARFs) for this study because, among solid-core and photonic bandgap fibers, ARFs have the advantages of low loss, low dispersion, low nonlinearity, high damage threshold, and high bandwidth. These properties might offer an opportunity to solve the issues that tie down the development of MCF, such as the difficulty in forming the supermode in coupled MCF and reducing crosstalk between different cores.

Recently, a multicore anti-resonant fiber (MC-ARF) has been proposed and manufactured [4]. However, only a few studies were conducted after this. Most of these studies focused on dual-core anti-resonant fibers (DC-ARFs), which are applied to beam splitters [5], couplers [6], and sensors [7]. No previous studies have linked MC-ARF with mode-division multiplexing. Only one previous study raised the number of cores to three; however, it did not expound on the transmission mechanism of such fibers [8]. The MCF design mentioned in this study is primarily aimed at fibers with three or more cores. The most practical ARF structure with the lowest confinement loss was proposed in 2021 [9], and its loss was 0.008 dB/km at 1550 nm. This fiber also achieved an ultra-high bandwidth of approximately 500 nm when the confinement loss was less than 0.05 dB/km. Although ARF has many advantages, it has many issues in MDM. In 2020, an ARF realized an MDM with seven modes while the confinement loss was as high as 0.6 dB/km at 850 nm, and the differential group delay between the fundamental mode (FM) and the higher-order mode (HOM) reached 2.5 ns/km [10]. In 2022, when working at 1550 nm, the confinement loss of an ARF increased to 0.18 dB/m at a bending radius of 0.75 cm [11]. These values are far higher than the best results accomplished by other fibers in the same application condition.

The transmission properties of an MDM system can be evaluated in several ways through numerical simulation. The three most significant aspects are a small enough loss (that can reduce the energy cost of long-distance transmission), a small enough chromatic dispersion (that can reduce the distortion of the signal), and a small enough differential group delay. However, both solid-core and hollow-core fibers cannot simultaneously keep CL, CD, and DGD in a relatively low state. In 2020, a three-core fiber achieved nearly zero chromatic dispersion; however, the differential group delay reached 2737 ps/km [12]. In the same year, an ARF controlled the chromatic dispersion below 10 ps/(km·nm) while the loss increased to 0.6 dB/km and the differential group delay increased to 250 ps/km [10]. If the standard SMF is considered as the benchmark [13], it is necessary to design a new fiber that can simultaneously achieve a loss lower than 0.2 dB/km and a chromatic dispersion lower than 16.7 ps/(km·nm). To reduce the complexity of the DSP algorithm, it should also achieve a differential group delay lower than 6 ps/km [14].

In this work, we propose a novel NMH-ARF with potential applications in MDM. We distribute the core in a triangular lattice. The modes in each core are strongly coupled through the air gap between the cores to form supermodes. The fiber structure can change from three-core weak coupling fibers to three-core and four-core strong coupling fibers under different parameters. The contents of this paper are organized as follows: The Section 2 focuses on the structure design of NMH-ARF. The Section 3 introduces the mode-coupling theory and transmission characteristics of this fiber. The Section 4 concludes the study.

## 2. Fiber Structure Design

According to the propagating principle of ARFs and the feasibility of future fabrication processes, we proposed the fiber structure in Figure 1. In ARFs, a structure with nested capillaries shows lower CL than a structure with simple capillaries [15]. Thus, the cladding in our design includes two layers of quartz-nested capillaries. In the outermost layer, $D_1$ denotes the diameters of the nested capillaries that are not conjunct with the adjacent capillaries. For easier reference, the names of each nested capillary in this paper are

replaced by their diameters, for instance, $D_1$ capillary. $D_3$ denotes the diameters of the nested capillaries in contact with the outer tube and $D_2$ capillary. The $D_1$ capillary is not in contact with the $D_2$ capillary, and the lowest distance between them is denoted by $l$. $D_2$ denotes the diameters of the inner cladding tubes that contact the $D_3$ capillary. The lowest width between each of the two neighboring $D_2$ capillaries is denoted by $g$, which is also known as the width of the coupling channel. $R$ denotes the inner radius of the outer tube. Meanwhile, there are three hollow cores (A, B, and C) and a central hollow core (O). The air regions of the four cores perform strong mode coupling between different cores through the coupling channel to produce supermodes. Three different kinds of nested capillaries have different structural designs. Each small tube in the nested capillaries has a diameter of $d_{11}$, $d_{22}$, and $d_{33}$, which corresponds to the diameters of $D_1$, $D_2$, and $D_3$, respectively. Each nested tube pair has three different ratios: $k_1$, $k_2$, and $k_3$.

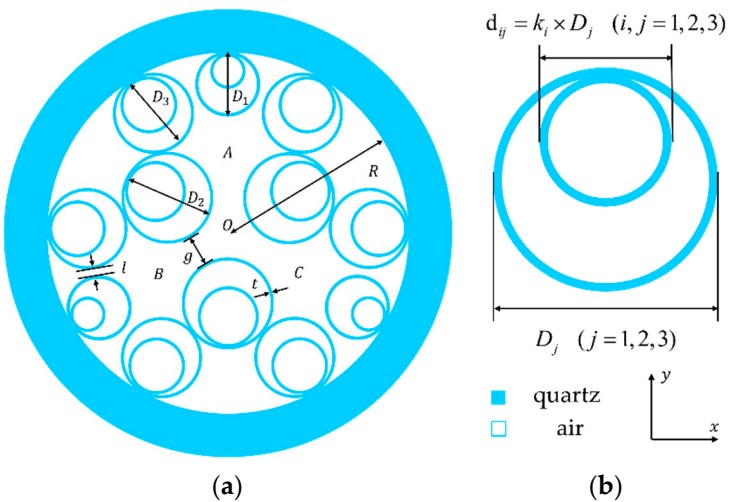

**Figure 1.** (**a**) Schematic diagram of the NMH-ARF structure; (**b**) Schematic diagram of the nested capillary structure.

In 2021, a double-ring negative-curvature hollow-core fiber was successfully fabricated by Y. Wang et al. [16], this fiber structure consisted of two layers of capillaries, and each inner capillary and the outer capillary are connected by only one node. Similarly, the preform of NMH-ARF can be constructed using the stacking technique. The nodes between the capillaries in Figure 1a ensure that the fiber designed in this study does not have a floating structure. Although the nodes in the structure will bring Fano resonance and increase CL, a small number of nodes can make the fiber fabrication easier because when the melted quartz structure is cooled during the fiber drawing process, the conjunctive tubes provide enough stress support to each other, which can improve the stability of the fiber structure. Further, the increase in CL can be reduced by adjusting the parameters, which means a few nodes can provide high feasibility for the fabrication of this fiber while ensuring the reduction in CL.

## 3. Results

### 3.1. Mode Analysis

The guiding mechanism of the fiber proposed in this study is the same as the ARROW theory in ARF [17]. The capillary wall works like a Fabry–Perot resonator, which can confine light in resonance conditions and reflect light in antiresonance conditions. If we set the capillary wall thickness $t$ to a fixed value, there is a conduction band of the fiber

between each of the two adjacent resonance wavelengths. The condition of resonance can be expressed as follows:

$$\lambda_{resonant} = \frac{2 \times t \times \sqrt{n_{cladding}^2 - n_{core}^2}}{m}, m = 1, 2, 3 \ldots, \tag{1}$$

where $n_{cladding}$ and $n_{core}$ are the refractive indices of the cladding and cores, respectively, and $m$ is any positive integer.

The full-vector finite element method can be applied in many fields such as optical fiber design [15], integrated optics devices [18,19], and sensing [20,21]. We used the full-vector finite element method to obtain the mode field distribution when the operating wavelength was 1.55 µm. A perfect matching layer (PML) surrounding the outer layer of the fiber can absorb the electromagnetic waves reaching the interface of the outer layer. During the establishment of the fiber model, we used PML to avoid abnormal reflection that might affect the accuracy of the simulation results. The fineness of the grid division in the ARF simulation will significantly impact the accuracy of the numerical simulation. We used a highly dense grid to improve this. The maximum size of the cell was set as $\lambda/6$ in the quartz area and $\lambda/4$ in the air area [15].

Figure 2 shows the mode field distribution of the fundamental supermode of the four-core strongly coupled anti-resonant fiber (FCS-ARF). The mode field of FCS-ARF is similar to that of the four-core solid-core strongly coupled fiber. There are four fundamental supermodes, each with two degenerate orthogonal polarized modes. In modes $LP_{01}^1$ and $LP_{01}^2$, we can observe that the energy is only distributed in the three cores (A, B, and C) while the four cores have an evident energy distribution in modes $LP_{01}^3$ and $LP_{01}^4$. These four modes can be reasonably explained by the mode-coupling theory. All the previous studies on MC-ARF were based on the mode-coupling theory [22]. This theory was originally proposed to solve the mode coupling caused by evanescent waves between the cores of solid-core multicore fibers. Theoretically, it can be assumed that each core can propagate independently and has its own propagation constant corresponding to its guided mode. When the pitch between the cores is large, the mode coupling between the cores is weak, which cannot produce supermodes, causing crosstalk between the cores. However, when the pitch is significantly small, the mode coupling strengthens and results in supermodes. In fact, the situation is slightly different in NMH-ARF. In terms of the principle, ARF and the total internal reflection fiber have significantly different light-guiding mechanisms. Because the total internal reflection fiber has guided modes, mode coupling between different cores can only be realized by evanescent waves. ARF does not support any guided modes except a leaky mode [23]. For an ARF surrounded by single-layer capillaries, the optical pulse tends to leak rather than propagate as the capillary pitch increases. Hence, CL is the dominant attenuation of this fiber, and the mode with a smaller CL is more stable in the fiber than the mode with a larger CL. This results in the biggest difference between multicore solid-core fibers and MC-ARFs. A significant criterion for the different kinds of coupling in MCF is the width between cores. If the core pitch of MCFs is large enough, the fiber transforms into an uncoupled fiber. However, the coupling method of NMH-ARF is not determined by the core pitch; it primarily depends on $g$, the width of the coupling channel. The mode in NMH-ARF tends to leak, which means the coupling will not disappear, and NMH-ARF must be a coupled MCF. This kind of coupling is illustrated in Figure 3. We can calculate the influence of the changing width of the coupling channel on CL if the parameters are set as shown in Table 1.

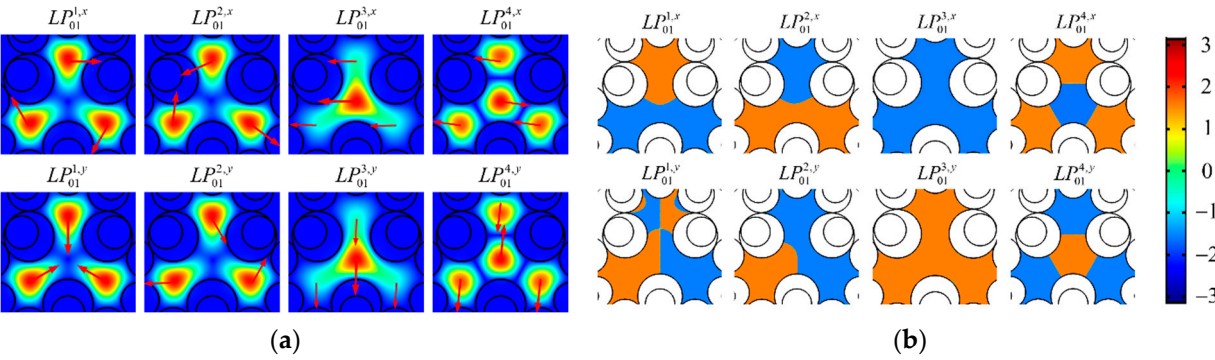

**Figure 2.** (**a**) Fundamental supermode electric field distribution of the four-core strongly coupled anti-resonant fiber, the direction of red arrows represents the direction of the electric field; (**b**) Fundamental supermode phase distribution of the four-core strongly coupled anti-resonant fiber and its color legend.

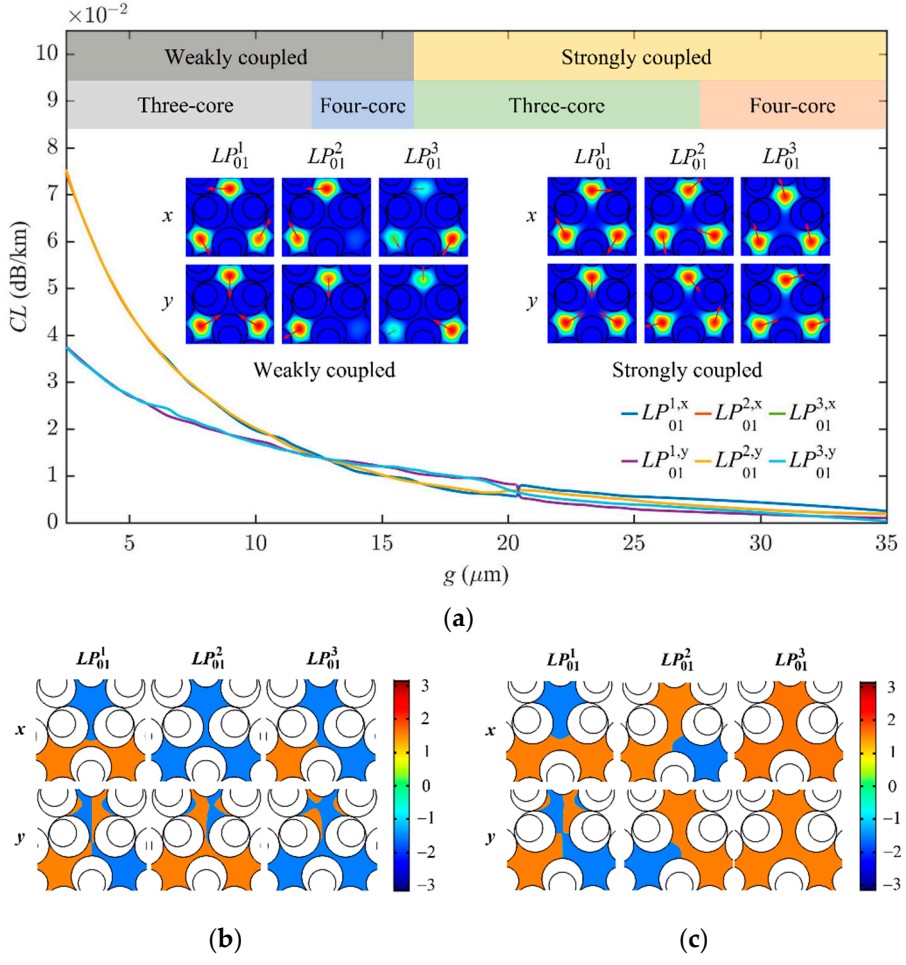

**Figure 3.** (**a**) Influence of various *g* on CL and the coupling characteristics, the direction of the red arrows represents the direction of the electric field; (**b**) Fundamental supermode ($LP_{01}^{1,x}$, $LP_{01}^{1,y}$, $LP_{01}^{2,x}$, $LP_{01}^{2,y}$, $LP_{01}^{3,x}$, and $LP_{01}^{3,y}$) phase distribution of the weakly coupled fiber and its color legend; (**c**) Fundamental supermode ($LP_{01}^{1,x}$, $LP_{01}^{1,y}$, $LP_{01}^{2,x}$, $LP_{01}^{2,y}$, $LP_{01}^{3,x}$, and $LP_{01}^{3,y}$) phase distribution of the strongly coupled fiber and its color legend.

**Table 1.** Parameter settings when calculating the influence of g on CL and the coupling characteristics.

| $R$ | $D_1$ | $D_2$ | $l$ | $t$ | $k_1$ | $k_2$ | $k_3$ |
|---|---|---|---|---|---|---|---|
| 125 μm | 44.98 μm | 63.02 μm | 4.7 μm | 0.52 μm | 0.52 | 0.66 | 0.7 |

$D_3$ is the function of $D_1$, $D_2$, $l$, and $g$. When 2.5 μm < $g$ < 35 μm, NMH-ARF switches between four different transmission characteristics based on the electric mode distribution. When 2.5 μm < $g$ < 12 μm, NMH-ARF transforms into a three-core weakly coupled anti-resonant fiber (TCW-ARF). When 12 μm < $g$ < 16.3 μm, NMH-ARF transforms into a four-core weakly coupled anti-resonant fiber (FCW-ARF). When 16.3 μm < $g$ < 25.7 μm, NMH-ARF transforms into a three-core strongly coupled anti-resonant fiber (TCS-ARF). Finally, when 25.7 μm < $g$ < 35 μm, NMH-ARF transforms into FCS-ARF. Since the linear polarization (LP) mode has two different polarizations, eight fundamental supermodes are found when four cores are coupled with each other, and six fundamental supermodes are excited when three cores are coupled with each other. The characteristics of the $LP_{01}^4$ mode change differently to the other six fundamental supermodes with the change in $g$. The CL and electric field distributions in the different mode coupling cases of the other six supermodes ($LP_{01}^{1,x}$, $LP_{01}^{1,y}$, $LP_{01}^{2,x}$, $LP_{01}^{2,y}$, $LP_{01}^{3,x}$, and $LP_{01}^{3,y}$) are depicted in Figure 3a. The $LP_{01}^4$ mode is discussed in Section 3.5 of this paper. With the increase in $g$, the confinement loss of the six modes mentioned before decreases gradually, and its magnitude decreases to $10^{-3}$.

The mechanism of this special mode coupling can be regarded as two equivalent processes, and these two processes can be described by the change in $g$. On the one hand, when $g = \sqrt{3}R - (2 + \sqrt{3})r_2$, the $D_2$ capillaries are close to the outer tube and the fiber can be seen as a single-core anti-resonant fiber with no coupling between the cores and all the energy distributed in the center. When $g$ is gradually reduced to approximately zero, the energy in the center is divided by the three $D_2$ capillaries and coupled to three gradually emerging cores (A, B, and C). Due to the longitudinal uniformity and transverse circular symmetry of the fiber structure, FMs and HOMs always maintain orthogonality and will not couple with each other in this slowly changing process. Therefore, except for the central core O, the mode distribution in the cores A, B, and C can be regarded as equal divisions of these optical energies. On the other hand, if $g$ is increased gradually, when $g$ is near zero, each hollow core (A, B, and C) can be regarded as a separated low-loss single-core anti-resonant fiber, as shown in Figure 4. The structure in Figure 4 is an independent part of the fiber in Figure 3, which indicates the change in CL in the fundamental mode in single-core ARFs when $g$ changes. In Figure 4a, this is derived as 2.5 μm < $g$ < 16.3 μm. The CL curves of the three individual cores show a flat trend that reaches 1.03 dB/km. When $g$ > 16.3 μm, CL increases exponentially in cores A, B, and C, and the mode leakage increases sharply. Similarly, in Figure 4b, the single central core O contains no guided mode when $g$ < 12 μm. After this, CL gradually decreases to a minimum value at $g$ = 16.3 μm. The mode leakage is always high during the change because the magnitude of CL is greater than $10^3$.

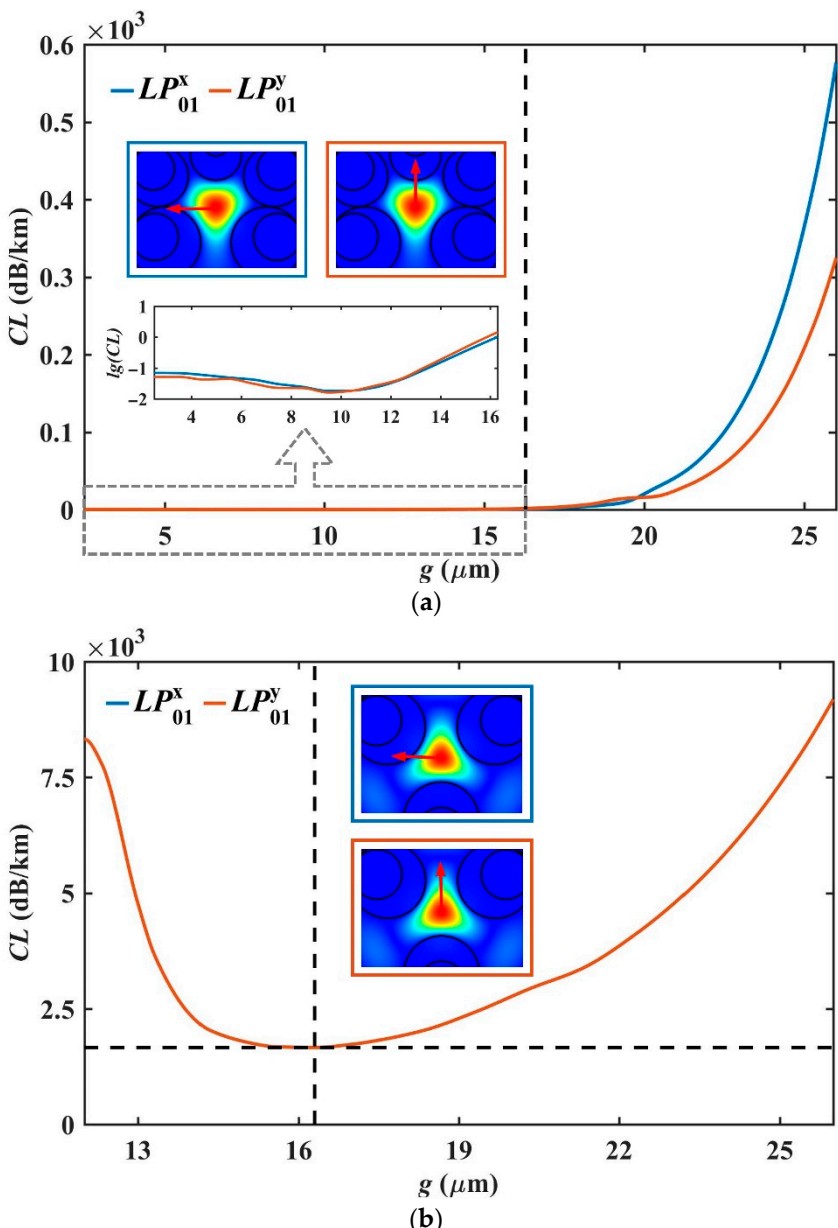

**Figure 4.** (**a**) Influence of various *g* on CL of three individual cores (A, B, and C) and the electric field distributions of x-polarization and y-polarization FM for the three individual cores, the direction of red arrows represents the direction of the electric field; (**b**) influence of various *g* on CL of the individual central core O and the electric field distributions of x-polarization and y-polarization FM for the individual central core, the direction of red arrows represents the direction of the electric field.

The results in Figure 3 match the results in Figure 4 well. When 2.5 μm < *g* < 12 μm, the guided mode in core O is cut off, the light is completely distributed in three other cores, and the three cores are weakly coupled through low mode leakage. In this case, the fiber transforms into TCW-ARF. When 12 μm < *g* < 16.3 μm, the guided mode starts to appear at core O, whereas the other cores can still be regarded as three low-loss ARFs. Thus, the four cores cannot have strong mode coupling, they are weakly coupled, and the TCW-ARF transforms into the FCW-ARF. When 16.3 μm < *g* < 25.7 μm, the leaky loss of the four cores increases sharply, and the coupling between the cores is enhanced. As a result, the supermodes are excited by strong mode coupling. Due to the light-guiding mechanism of ARFs, single-mode transmission can be realized when FMs have a lower confinement loss than HOMs. Therefore, the confinement loss of the $LP_{01}^4$ mode can be much higher than

the other six fundamental supermodes by adjusting the structure, from which FCW-ARF can become TCS-ARF. Finally, when 25.7 μm < $g$ < 35 μm, the coupling becomes strong in every core, the loss of the $LP_{01}^4$ mode is almost the same as the other six fundamental supermodes, and the fiber appears as FCS-ARF. When the $D_2$ capillaries are close to the outer tube, the four individual cores disappear, and the air regions of each core merge into one hollow core. The fiber turns into a single-core ARF. The two converse variations of $g$ suggest that FMs and HOMs are orthogonal, and the fundamental supermodes is only excited by the coupling of FMs in each individual core. Consequently, we can derive an equation that describes the mode coupling corresponding to this structure, assuming that the total electronic field distribution across the fiber cross-section is expressed as [24]:

$$E_{(x,y,z)}^{supermode} = \sum_{i=1}^{4} A_i(z) E_i(x,y) exp(j\beta_i z), \tag{2}$$

where $A_i(z)$ is the amplitude of FMs in each core, $E_i(x,y)$ is the original transverse field of electronics for each core, and $\beta_i$ is the FM propagation constant of each core. If we set $a_i(z) = A_i(z)exp(j\beta_i z)$, we can derive the following mode coupling equation:

$$\frac{da_i(z)}{dz} = j\beta_i a_i(z) + \sum_q j\kappa_{iq} a_q(z), i \neq q, i = 0, 1, 2, 3, \tag{3}$$

where $i$ and $q$ are the number of cores, and $\kappa_{iq}$ is the coupling coefficient between cores $i$ and $q$. Thus, this equation can be simplified to:

$$\frac{da_i(z)}{dz} = jKa_i(z), K = \begin{bmatrix} \beta_1 & \kappa_{12} & \kappa_{13} & \kappa_{10} \\ \kappa_{21} & \beta_2 & \kappa_{23} & \kappa_{20} \\ \kappa_{31} & \kappa_{32} & \beta_3 & \kappa_{30} \\ \kappa_{01} & \kappa_{02} & \kappa_{03} & \beta_0 \end{bmatrix}, \tag{4}$$

where $K$ is the coupling matrix. Because of the central symmetry of this fiber, the FM propagation constants and the coupling coefficients of each core are the same as constants $\beta$ and $\kappa$, respectively:

$$\begin{aligned} \beta_1 = \beta_2 = \beta_3 = \beta, \\ \kappa_{12} = \kappa_{13} = \kappa_{21} = \kappa_{23} = \kappa_{31} = \kappa_{32} = \kappa \end{aligned} \tag{5}$$

Although the structure of core O is different from the other three cores, the coupling coefficients between core O and each of the other cores are the same. Therefore, this coupling coefficient is set as $\eta$. The propagation constant of the fundamental mode in core O is set as $\alpha$:

$$\beta_0 = \alpha, \kappa_{10} = \kappa_{20} = \kappa_{30} = \kappa_{01} = \kappa_{02} = \kappa_{03} = \eta, \tag{6}$$

Substituting Equations (5) and (6) into Equation (4) yields a differential equation as follows:

$$\frac{da_i(z)}{dz} = j \begin{bmatrix} \beta & \kappa & \kappa & \eta \\ \kappa & \beta & \kappa & \eta \\ \kappa & \kappa & \beta & \eta \\ \eta & \eta & \eta & \alpha \end{bmatrix} a_i(z), \tag{7}$$

The coupling matrix can be diagonalized, and the reversible matrix $Q$ and $Q^{-1}$ can help derive the following equation:

$$K = Q\Lambda Q^{-1}, \Lambda = \begin{bmatrix} \gamma_1 & & & \\ & \gamma_2 & & \\ & & \gamma_3 & \\ & & & \gamma_0 \end{bmatrix}, \tag{8}$$

where $\Lambda$ is the diagonal matrix corresponding to the coupling matrix, and $\gamma_i$ ($i$ = 0,1,2,3) represents the eigenvalues corresponding to the secular equation determined by the coupling matrix $K$. These four eigenvalues correspond to the four supermodes, indicating the propagation constants of the four supermodes [12]:

$$j(\gamma I - K) \begin{bmatrix} \chi_1 \\ \chi_2 \\ \chi_3 \\ \chi_0 \end{bmatrix} = 0, \tag{9}$$

To make the secular equation solvable, the coupling determinant must be zero:

$$(\kappa - \beta + \gamma)^2 \left( 3\eta^2 - (\alpha - \gamma)(\beta - \gamma + 2\kappa) \right) = 0, \tag{10}$$

The solution of this equation can correspond to the numerical simulation results of NMH-ARF well:

$$\begin{array}{ll} \textit{Weakly \quad coupled} \\ \eta \to 0 \end{array} \left\{ \begin{array}{l} \gamma_1 \to \beta + 2\kappa \\ \gamma_2 = \gamma_3 = \beta - \kappa \\ \gamma_4 \to \alpha \end{array} \right. \tag{11}$$
$$\begin{array}{ll} \textit{Strongly \quad coupled} \\ \eta \gg 0 \end{array} \left\{ \begin{array}{l} \gamma_1 = \frac{\alpha+\delta}{2} + \frac{1}{2}\sqrt{(\alpha-\delta)^2 + 12\eta^2}, \\ \gamma_2 = \gamma_3 = \beta - \kappa \\ \gamma_4 = \frac{\alpha+\delta}{2} - \frac{1}{2}\sqrt{(\alpha-\delta)^2 + 12\eta^2} \end{array} \right.$$

where $\delta = \beta + 2\kappa$.

Equation (11) shows that when $\eta$ is significantly large, Equation (10) has four nontrivial solutions. The same solutions $\gamma_2$ and $\gamma_3$ represent the two supermodes' propagation constants modulated by cores A, B, and C when mode coupling, and both $\gamma_2$ and $\gamma_3$ are independent of the central core. Moreover, in the $LP_{01}^1$ and $LP_{01}^2$ modes, we discovered that there is no electric field distribution around the central point of this fiber. Thus, the $\gamma_2$ and $\gamma_3$ solutions correspond to the $LP_{01}^1$ and $LP_{01}^2$ modes, respectively. The simultaneous occurrence of $\alpha$, $\beta$, $\eta$, and $\kappa$ in the $\gamma_1$ and $\gamma_4$ solutions indicates that the four cores modulate the other two supermodes' propagation constants together. Further, in the $LP_{01}^3$ and $LP_{01}^4$ modes, the electric field distribution is not zero near the center point of the structure. Therefore, the $\gamma_1$ and $\gamma_4$ solutions correspond to the $LP_{01}^3$ and $LP_{01}^4$ modes, respectively. Meanwhile, $\eta \gg 0$ indicates that the coupling between core O and the other cores is significantly strong. As a result, when $\eta \gg 0$, the propagation constant solution corresponds to the strongly coupled fiber. When $\eta$ approaches zero, $\gamma_2$ and $\gamma_3$ remain unchanged, $\gamma_1$ approaches $\beta + 2\kappa$, and $\gamma_4$ approaches $\alpha$, which shows that the coupling between the four cores is weakened, and the fiber shifts into FCW-ARF. When $\alpha = 0$, the mode in core O cuts off, and the corresponding coupling coefficient $\eta$ becomes zero such that $\gamma_4$ does not exist, and the other three core propagation constants are only related to $\beta$ and $\kappa$; thus, the fiber transforms to TCW-ARF.

Based on the above analysis, Equation (8) is substituted into Equation (7). The differential equation can be solved using the matrix exponential method [25] according to the obtained eigenvector. The solution of Equation (7) is:

$$\begin{bmatrix} a_1(z) \\ a_2(z) \\ a_3(z) \\ a_0(z) \end{bmatrix} = Q \exp(z\Lambda) Q^{-1} \begin{bmatrix} a_1(0) \\ a_2(0) \\ a_3(0) \\ a_0(0) \end{bmatrix}, \tag{12}$$

where $\alpha_i$ (0) ($i$ = 0,1,2,3) is determined by the initial conditions of the input light of each fiber core. The mode distribution of the four fundamental supermodes can be derived

by substituting Equation (12) into Equation (2). Each supermode has two degenerate polarizations in the x and y directions, from which eight modes are obtained.

### 3.2. Characteristics of CL

The previous section discussed the influence of the variable $g$ on CL of different modes and the mode-coupling theory used in this paper. In our study, the parameters that affect the transmission performance of the fiber are $r_1$, $r_2$, $l$, $g$, $t$, $k_1$, $k_2$, and $k_3$, in which $r_1$ and $r_2$ are the radii of the $D_1$ capillaries and $D_2$ capillaries, respectively. The modes in ARFs tend to leak out from the core rather than be confined, and the modes that can be transmitted are propagated in the air rather than quartz. Thus, confinement loss is the primary loss of this kind of fiber. When working at a wavelength of $\lambda_{work}$, the expression of CL is as follows [26]:

$$L_c = -\frac{20}{\ln 10} \frac{2\pi}{\lambda_{work}} \mathrm{Im}\left(n_{eff}\right), \tag{13}$$

where $\mathrm{Im}(n_{eff})$ is the imaginary part of the effective refractive index of the supermodes. To identify the parameters with the lowest CL, this section outlines the numerical simulation results of each parameter. When $g$ = 20.32 μm, Figures 5 and 6 show the influence of the change in the different parameters on CL. When each parameter varies independently, the values of the other parameters are also independent and fixed at the same parameter settings as in Section 2. Each variable in the figure corresponds to the maximum and minimum CL of the six fundamental supermodes. We observe that the curves of $r_1$ and $r_2$ almost have the same trend. The maximum loss $CL_{max}$ first decreases to an extremely low point with the increase in $r_1$ and $r_2$ and then increases. However, the minimum loss $CL_{min}$ decreases as the radii increase and tends to be stable. This is because the change in $r_1$ and $r_2$ is actually the size of the change in the cores A, B, and C. As $D_3$ is the function of $r_1$, $r_2$, $l$, and $g$, the three cores become significantly small if the radius is extremely large or small. The coupling between the four cores is weakened, and most of the energy is confined to the center. This makes the structure tend to be a single-core ARF, which indicates an increasing CL of the supermodes and a decreasing number of modes.

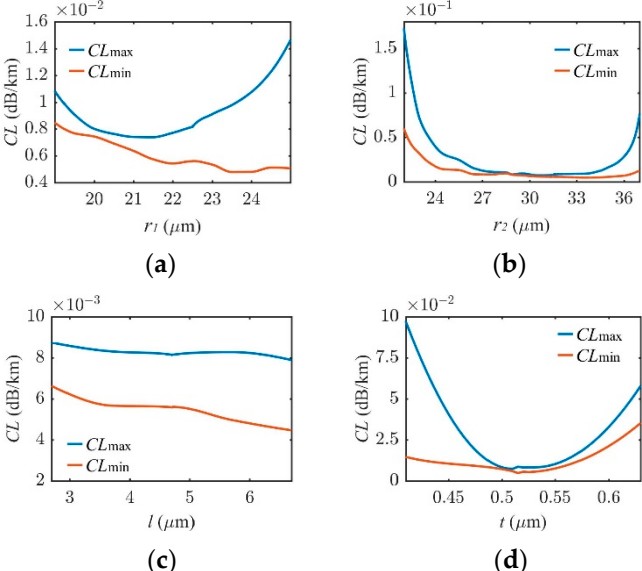

**Figure 5.** (**a**) Change in CL with $r_1$; (**b**) Change in CL with $r_2$; (**c**) Change in CL with $l$; (**d**) Change in CL with $t$.

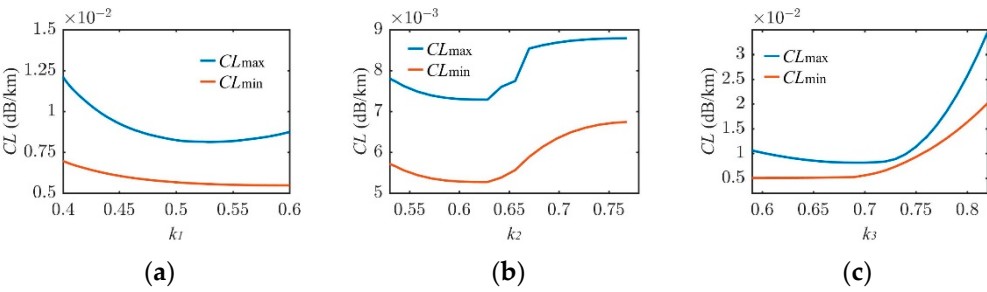

**Figure 6.** (**a**) Change in CL with $k_1$; (**b**) Change in CL with $k_2$; (**c**) Change in CL with $k_3$.

The capillary wall thickness *t* is the most important parameter in ARFs. In Figure 5d, with an increase in *t*, $CL_{max}$ and $CL_{min}$ first decrease and then increase, owing to the ARROW theory in ARF [17]. However, the curves of the change in CL with *l* are relatively flat because the change in *l* has less of an effect on the shape of the core when CL is kept below 0.009 dB/km.

The small tube in the nested capillaries enhances the confinement of the core modes [16]. Figure 6 depicts how the ratio $k_1$, $k_2$, and $k_3$ changes CL. Several previous studies have computed this [27]. However, in these studies, the nested capillaries at different positions used the same proportion coefficient. Our study found that the nested capillaries at different positions reached the minimum CL at different ratios. Figure 6a–c show different trends. The curves of the change in CL with $k_2$ appear to be periodic. At the maximum or minimum $k_2$, the value of CL tends to be the same. There is no such property in the other two parameters, in which the change in CL with $k_1$ is relatively flat; however, the change in CL with $k_3$ is relatively large. In terms of the tube position, the nested capillaries near the coupling channel directly affect the coupling of each core; however, the nested capillaries away from the coupling channel only directly affect the modes of cores A, B, and C.

*3.3. Characteristics of Dispersion*

The chromatic dispersion includes waveguide dispersion and material dispersion. In ARFs, when an optical pulse passes through the structure, the energy is distributed to different modes. Different shapes of the structure have different group velocity responses to different frequency components in a certain mode, resulting in different latency at the output port and waveguide dispersion. Similarly, different frequency components passing through different materials produce different group velocities, which results in group delays at the end of the fiber, resulting in the material dispersion and the broadening of the optical pulse. If the refractive index of the quartz is expressed by the Sellmeier equation, the pulse broadening at different wavelengths can be described by the chromatic dispersion coefficient D as follows [12]:

$$D(\lambda) = -\frac{\lambda}{c}\frac{d^2\text{Re}\left(n_{eff}\right)}{d\lambda^2},\tag{14}$$

where *c* is the speed of light in vacuum, and $\text{Re}(n_{eff})$ is the real part of the effective refractive index of supermodes.

CD is not the only reference index for the MDM system. When multiple channels are able to simultaneously transmit in the fiber, different modes have different paths under the same wavelength. Thus, they have different axial speeds, resulting in inter-mode dispersion. In the multicore fiber, the dispersion between any two modes *i* and *j* is usually described by the differential mode group delay as follows [2]:

$$DGD_{ij} = \frac{n_{eff}^{(i)} - n_{eff}^{(j)}}{c} - \frac{\lambda}{c}\left(\frac{dn_{eff}^{(i)}}{d\lambda} - \frac{dn_{eff}^{(j)}}{d\lambda}\right),\tag{15}$$

where $n_{eff}^{(i)}$ and $n_{eff}^{(j)}$ are the effective refractive indices of the $i$ and $j$ mode, respectively.

Adjusting the dispersion in NMH-ARF is easy. We obtain low dispersion at different parameters, making the application of the fiber more flexible. Figures 7 and 8 describe the relationship between $g$, $l$, $r_1$, and $r_2$ with the dispersion coefficient and differential group delay at a wavelength of 1.55 μm with the parameter set at the values mentioned in Section 2. The adjustment of these four parameters does not have a significant impact on the dispersion coefficient; however, it has a significant impact on DGD. Each DGD$_{max}$ curve has two zero points, and the minimum DGD of the same parameter is always around zero.

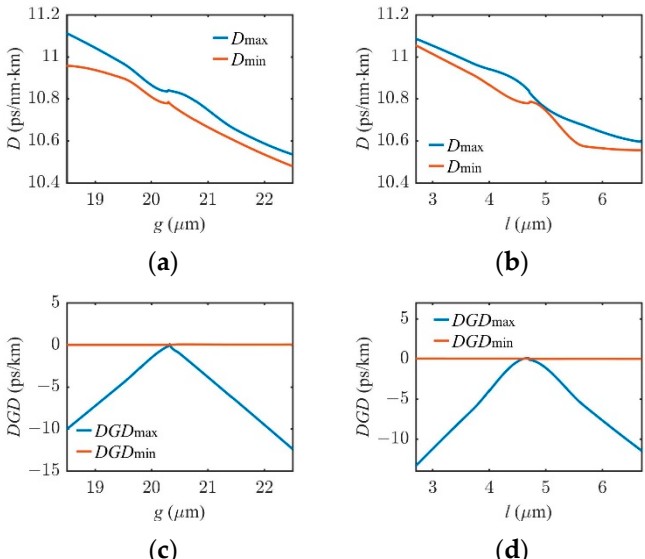

**Figure 7.** (**a**) Change in D with $g$; (**b**) Change in D with $l$; (**c**) Change in DGD with $g$; (**d**) Change in DGD with $l$.

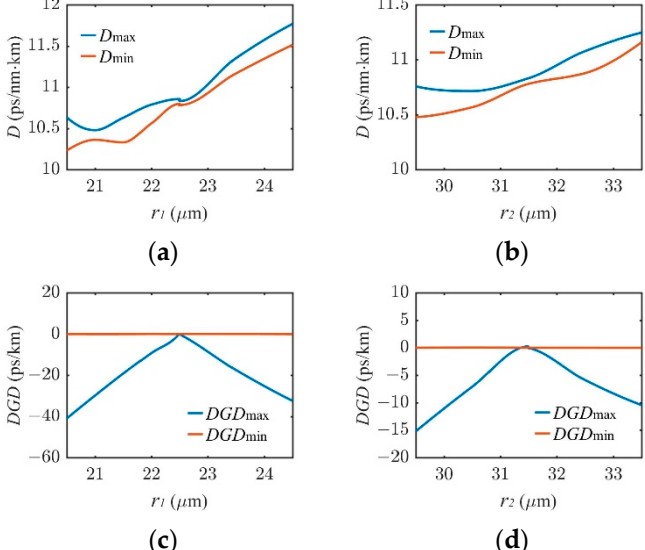

**Figure 8.** (**a**) Change in D with $r_1$; (**b**) Change in D with $r_2$; (**c**) Change in DGD with $r_1$; (**d**) Change in DGD with $r_2$.

The influence of the wall thickness on the dispersion differs from the other parameters. In Figure 9, within a short range of variation, the adjustment of $t$ makes the dispersion coefficient and differential group delay reach zero. When $t < 0.52$ μm, the curves of D and DGD are relatively flat. When $t > 0.52$ μm, the dispersion curves change quickly.

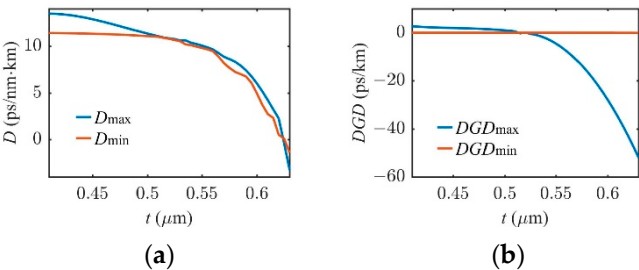

**Figure 9.** (**a**) Change in D with *t*; (**b**) Change in DGD with *t*.

Compared to the other parameters, the ratio $k_1$, $k_2$, and $k_3$ has a different influence on the characteristics of the supermode dispersion. In Figure 10, within a large range of variation, the $DGD_{max}$ curves of these three parameters change almost linearly, with a zero point in them. Moreover, the variation of D and $DGD_{min}$ is relatively flat. This provides this fiber structure with the most flexible dispersion adjustment characteristics. $k_1$, $k_2$, and $k_3$ can not only adjust the differential group delay to zero but also keep the confinement loss super low compared to the other parameters when the dispersion coefficient is around 10.78 ps/(km·nm) under the parameter settings of Table 2.

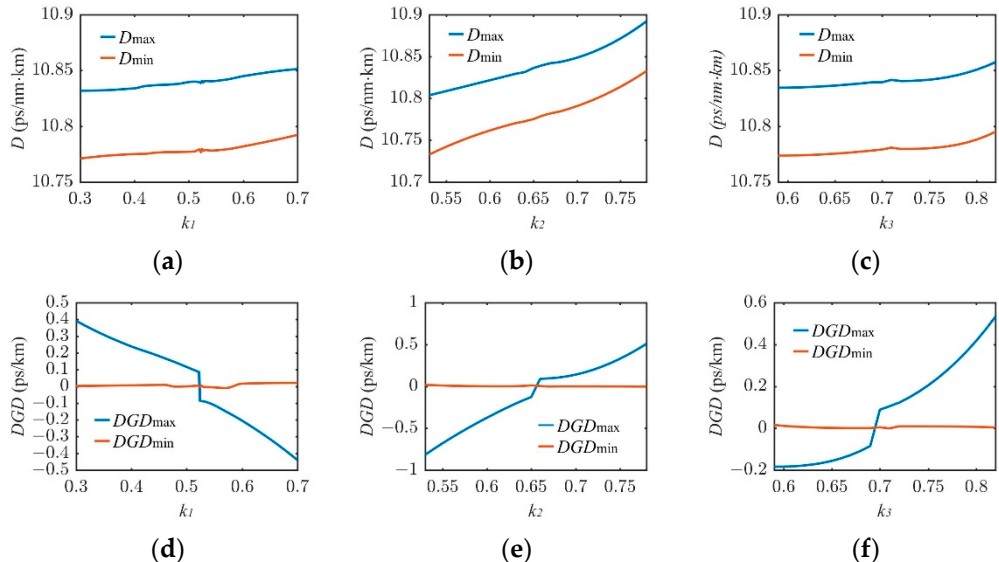

**Figure 10.** (**a**) Change in D with $k_1$; (**b**) Change in D with $k_2$; (**c**) Change in D with $k_3$; (**d**) Change in DGD with $k_1$; (**e**) Change in DGD with $k_2$; (**f**) Change in DGD with $k_3$.

**Table 2.** Parameter settings when reaching the minimum of the maximum dispersion coefficient and differential group delay.

| $g$ | $r_1$ | $r_2$ | $l$ | $t$ | $k_1$ | $k_2$ | $k_3$ |
|---|---|---|---|---|---|---|---|
| 20.32 μm | 22.49 μm | 31.51 μm | 4.7 μm | 0.52 μm | 0.522 | 0.66 | 0.7 |

The maximum DGD of the six fundamental supermodes is reduced to 0.085 ps/km and the maximum CL is reduced to 0.008 dB/km. According to our investigation, this is currently the lowest loss and DGD that can be achieved in mode-division multiplexing.

### 3.4. Bending Resistance

To make the fiber design in this study more feasible, this section explores the characteristics of bending loss. Although ARF has a lower cost of fabrication and lower loss than solid-core fiber, it still has the issue of a large bending loss. The issue of reducing the bending loss of ARF has remained for years and has always been a research hotspot. In a

computational simulation, the bending radius of the fiber often connects to the change in the refractive index:

$$n_{bend}(x) = n_{caldding}\left(1 + \frac{x}{R_{bend}}\right), \tag{16}$$

where $n_{bend}(x)$ is the refractive index distribution on the *x*-axis after fiber bending, and $R_{bend}$ is the corresponding bending radius [15]. The origin of the coordinate axis is at the center of the fiber in Figure 11.

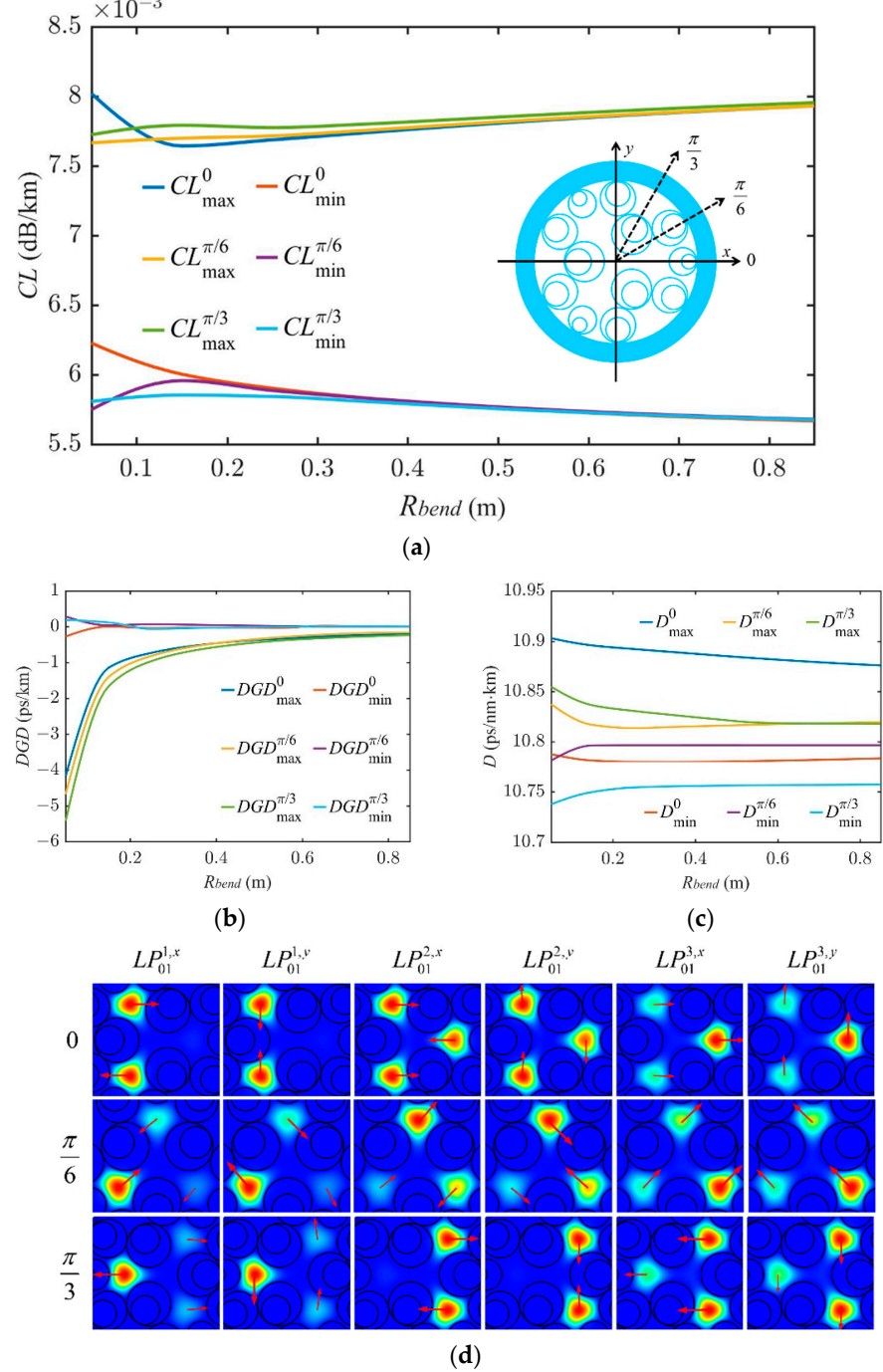

**Figure 11.** (**a**) Change in CL with the bending radius; (**b**) Change in DGD with the bending radius; (**c**) Change in D with the bending radius; (**d**) Electric field distribution of six fundamental supermodes when the bending radius is 0.05 m at three angle conditions (0, $\pi/6$, and $\pi/3$), the direction of red arrows represents the direction of the electric field.

Equation (16) shows that the propagation of light in a bent fiber is equivalent to the conformal mapping of the refractive index of a dielectric [27]. Considering that the bending loss is caused by the bending of quartz and the bending of the air–dielectric boundary in the hollow core fiber, the transformation of this equation is aimed at the quartz and air–quartz interface. As the fiber structure in this study has central and axial symmetries, the loss, dispersion coefficient, and DGD of the fiber under different bending radii are calculated from three angles relative to the *x*-axis: 0, $\pi/6$, and $\pi/3$. We observe that when the bending radius changes from 0.05 to 0.85 m, the changing amplitude of the confinement loss and dispersion coefficient corresponding to the three angles is significantly small. The curves of DGD change rapidly when the bending radius changes from 0.05 to 0.2 m. After this, the curves become stable. This is because the bending structure has little impact on the coupling channel. Since the modes that are transmitted in the fiber are supermodes, the change in the structure has little impact on the strong coupling of the three cores. Therefore, the optical fiber structure in this study provides good bending resistance and has great potential for application in ultra-long-distance communication.

*3.5. Selective Transmission of Supermodes*

This section explains why there is a particular case of TCS-ARF in the process of four-core coupling. The higher-order mode extinction ratio (HOMER) is defined as the ratio of the minimum loss of higher-order modes to the maximum loss of fundamental modes [28]. Because the refractive index of the core is much smaller than that of the cladding, the hollow-core fiber has no mode cut-off, and the single-mode operation characteristics can only be achieved by obtaining a larger HOMER at a suitable transmission distance. There is no standard for HOMER. As long as the loss of FMs is much lower than that of higher-order modes or cladding modes, the single-mode operation can always be realized at an appropriate transmission distance. In 2010, a photonic crystal fiber with a negative curvature successfully realized single-mode transmission in the range of $8 \leq \text{HOMER} \leq 26$ [29], which means if HOMER > 8, the fiber can realize single-mode operation.

When the four cores are strongly coupled, due to the weak propagation capacity and high mode leakage of core O, when the parameters are satisfied, CL of $\text{LP}_{01}^4$ mode is much greater than the other six modes to realize $\text{LP}_{01}^4$ mode extinction during long-distance transmission. A new parameter—center mode extinction ratio (CER)—is defined as:

$$CER = \frac{Lowest\ loss\ of\ \text{central mode}}{Highest\ loss\ of\ \text{fundamental mode}}, \tag{17}$$

When CER is significantly high, the energy in the central core is completely coupled to the other three cores, and the coupling between cores A, B, and C becomes significantly stronger. At this time, it is regarded as TCS-ARF. Figures 4 and 12 are well matched. Figure 12 shows that when 12 μm < *g* < 16.3 μm, CER decreases from 4202.23 to 11.87. After this, CER begins to increase and reaches a maximum value of 61.82 in the dotted line area of the picture. Meanwhile, in Figure 4b, the loss of the central mode decreases and reaches the minimum at *g* = 16.3 μm. When *g* > 16.3 μm, the loss of the central mode begins to gradually increase. The three-core strong coupling state does not appear suddenly. However, the minimum CER value in the three-core strong coupling area is higher than 3.

Under the parameters mentioned in Section 2, we calculate the curves of the confinement loss and dispersion coefficient in TCS-ARF at wavelengths from 1.25 to 1.75 μm. In Figure 13a, the maximum loss of the six supermodes fluctuates slightly between 0.024 and 0.007 dB/km, and the curves reach the minimum value of 0.007 dB/km at 1.449 μm. In the whole O, E, S, C, L, and U bands, the confinement loss is always lower than 0.029 dB/km. In the same range of wavelengths, the maximum dispersion coefficient of this fiber varies between −7.037 and 13.283 ps/(km·nm), which rapidly changes before 1.5 μm. Further, the curve is relatively flat in the whole band of the wavelength, and all dispersion coefficients satisfy the requirements of MDM. In Figure 13c, the maximum of DGD varies significantly between 1.25 μm < *λ* < 1.4 μm, and the absolute value of DGD is higher than

4.9 ps/(km·nm). In the other bands, DGD changes smoothly, the minimum value can reach zero, and the value fluctuates between −4.9 and 0.089 ps/(km·nm).

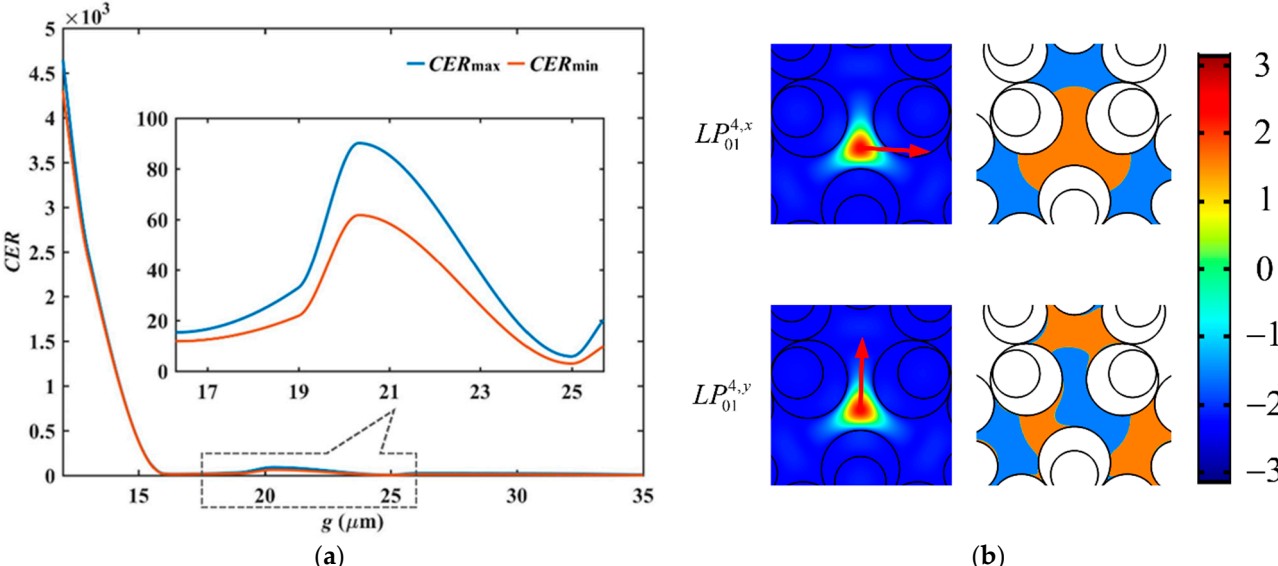

(a)　　　　　　　　　(b)

**Figure 12.** (**a**) Change in the central mode extinction ratio with $g$; (**b**) Electric field distribution (left) and phase distribution (right) of the fundamental supermode ($LP_{01}^{4,x}$, $LP_{01}^{4,y}$) for the three-core strongly coupled fiber. The color legend is for the phase distributions, the direction of red arrows represents the direction of the electric field.

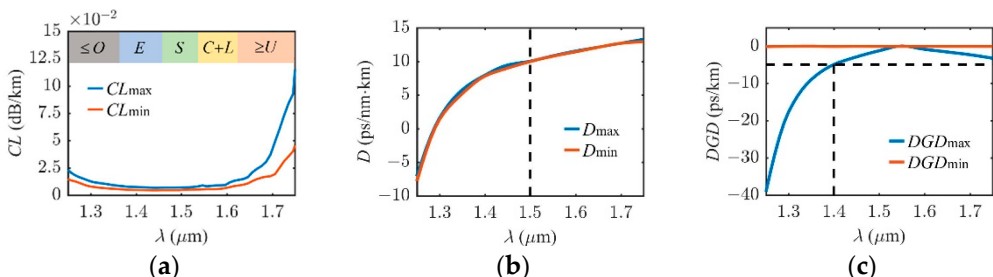

(a)　　　　　　　　　(b)　　　　　　　　　(c)

**Figure 13.** (**a**) CL of the change in TCS-ARF with $\lambda$. (**b**) D of the change in TCS-ARF with $\lambda$. (**c**) DGD of the change in TCS-ARF with $\lambda$.

In addition to CER, HOMER also needs to be considered in this study. Figure 14 shows the variation of HOMER of two higher-order modes $LP_{11}^a$ and $LP_{11}^b$.

In the whole O, E, S, C, L, and U bands, HOMER is always higher than 12. When $\lambda$ = 1.55 μm, HOMER of $LP_{11}^a$ is 57.42 and HOMER of $LP_{11}^b$ is 284.35, which indicates that the six fundamental supermodes can achieve low dispersion and low loss transmission at wavelengths from 1.4 to 1.75 μm.

Although Figures 12 and 13 show that the higher-order modes and the central mode will become extinct during long-distance transmission, a higher extinction ratio is required when it comes to a medium to short distance. To make an adjustment, we add an inscribed capillary tube between the small tube and the large tube in the $D_2$ capillaries. The nodes at the three points closest to the central core can result in Fano resonance in core O, and the central mode $LP_{01}^4$ can leak out through Fano resonance. There are also new nodes between the added capillary and the small tube in the $D_2$ capillaries. Because higher-order modes are easier to be coupled to the clad capillary [15], such nodes in nested tubes can also result in Fano resonance affecting the higher-order modes. The new added nodes have little effect on the coupling of the other three cores. As a result, CER and HOMER increase.

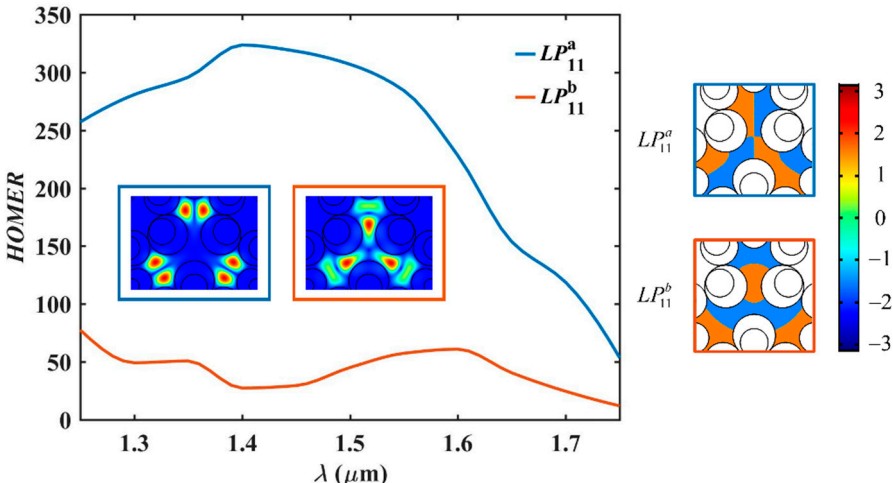

**Figure 14.** The change in HOMER with $\lambda$. The two most representative electric field distributions of the high-order supermodes $LP_{11}^a$ and $LP_{11}^b$ are on the left, and their corresponding phase distributions are on the right. The color legend is for the phase distributions.

Figure 15b shows that in the structure of Figure 15a, HOMER and CER are adjusted by $k_2$ at a wavelength of 1.55 μm. Since there are two nodes between the inscribed and nested tubes, and the center points of both capillaries are in the same line passing through the center point of the fiber, $k_2$ adjusts the size of the inscribed tubes. These curves show that different $k_2$ significantly changes the supermode selection characteristics of the fiber. When $k_2 > 0.4$, CER is higher than 116.1. When $k_2 = 0.4$, HOMER reaches the maximum value of 190.8. The maximum confinement loss of the fundamental supermodes in this structure is 0.013 dB/km, the maximum dispersion coefficient is 11.06 ps/(km·nm), and the maximum DGD is 1.03 ps/km. Therefore, inscribed tubes do not significantly impact the loss and dispersion, and mode selection is realized under the condition of ensuring low loss and low dispersion of the supermodes.

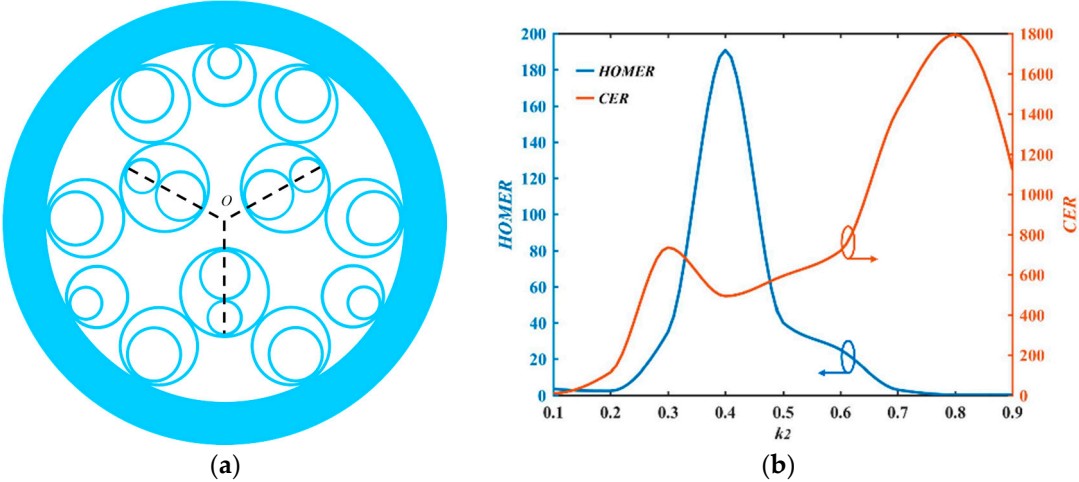

(**a**)         (**b**)

**Figure 15.** (**a**) The previous structure with the addition of a tube; (**b**) the change in HOMER and CER with $k_2$.

In addition, the dispersion of this fiber structure can easily reach the zero point through many parameter adjustments. To illustrate that the addition of inscribed tubes does not affect the dispersion adjustment characteristics, an example of the change in the characteristics with $k_1$ is selected to analyze such properties in the new structure. The trend of the curves in Figure 16 is the same as it in Sections 3.2 and 3.3. DGD can still reach zero by adjusting the parameters. In particular, when $k_1 = 0.7$, the maximum confinement loss

is 0.017 dB/km, the maximum dispersion coefficient is still 11.06 ps/(km·nm), and the maximum DGD is reduced to $-0.526$ ps/km. Meanwhile, HOMER is 144.2 and CER is 372, which significantly achieves mode selection. As a result, this fiber can realize MDM of six supermodes.

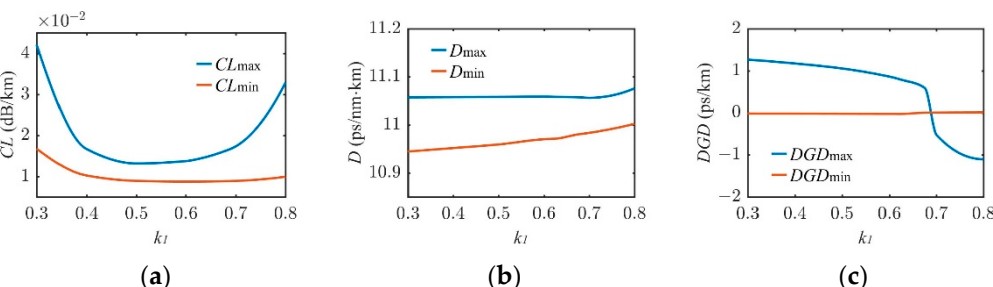

**Figure 16.** (**a**) Change in CL in the new structure with $k_1$; (**b**) Change in D in the new structure with $k_1$; (**c**) Change in DGD in the new structure with $k_1$.

## 4. Conclusions

In this study, we proposed an optical fiber structure which has great potential applications in MDM. We theoretically explained the special mode coupling characteristics in this fiber. Formulas were used to show the changes between strongly and weakly coupled fibers, three-core, and four-core fibers. Using the full-vector finite-element method, we calculated the confinement loss, dispersion coefficient, and differential group delay of NMH-ARF under different parameter settings and illustrated the bending resistance and mode extinction characteristics of the fiber. At a wavelength of 1.55 μm, when we set the parameters to $D_1 = 44.98$ μm, $D_2 = 63.02$ μm, $g = 20.32$ μm, $l = 4.7$ μm, $t = 0.52$ μm, $k_1 = 0.52$, $k_2 = 0.66$, and $k_3 = 0.7$, the maximum differential group delay of the different modes was reduced to 0.085 ps/km, and the maximum confinement loss was reduced to 0.008 dB/km. By changing the structure, HOMER and CER were increased to 190.8 and 372, respectively. As a result, the three key indicators of MDM were significantly reduced to low levels in the same fiber. However, the manufacturing is challenging, which will be explored in future experiments.

**Author Contributions:** Conceptualization, Z.L., Z.H. and G.Z.; methodology, Z.L. and J.L.; investigation, Z.L. and C.X.; resources, J.L., C.X., Z.H. and G.Z.; data curation, Z.L. and G.Z.; writing—original draft preparation, Z.L.; writing—review and editing, Z.L., Z.H. and G.Z.; supervision, Z.H. and G.Z.; project administration, Z.L.; funding acquisition, Z.H. and G.Z. All authors have read and agreed to the published version of the manuscript.

**Funding:** This work was supported by the National Natural Science Foundation of China (61935007).

**Institutional Review Board Statement:** Not applicable.

**Informed Consent Statement:** Not applicable.

**Data Availability Statement:** All data are provided in full in the results section of this paper.

**Acknowledgments:** The authors would like to thank their supervisors and colleagues who have helped a lot with the research.

**Conflicts of Interest:** The authors declare no conflict of interest.

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
