# Peer review of "Supermode Characteristics of Nested Multiple Hollow-Core Anti-Resonant Fibers"

_photonics, doi:10.3390/photonics9110816_

Round 1

Reviewer 1 Report

The authors proposed and numerically investigated hollow-core anti-resonant fiber construction supporting several supermode which may be interesting for mode-division multiplexing (MDM). Detailed numerical results obtained in the framework of FEM simulations are demonstrated. The optimization of fiber parameters is presented. The results are interesting. But I have the following comments and questions listed below.  

1) The authors claimed that the proposed fiber construction has significant potential for applications in MDM. However, the considered construction can support up to 4 supermodes (4x2 with allowance for two polarizations). What is about scalability? Is it possible to generalized the idea and modified the fiber construction for larger number of channels?

2) It is not entirely clear what is shown in Fig. 2, inserts in Fig. 14, and other figures signed "field distribution". Is it |E|? If yes, what is about phases? For MCF supermodes, phases may differ by pi in neighboring cores or may be the same. For example, the phase in the central core may differ by pi from phases in “surrounding” cores. It would be useful to add figures containing information about phases (and/or give the corresponding explanation in the text).

3) In several plots, there are “jumps” on curve which seem strange (for example, Fig. 5b, orange curve, r2~23; Fig 6b, k2~0.65; Fig 13a, lambda>1.6; and so on). Are they results of numerical errors or could there be another explanation? If these are numerical errors, how can this issue be solved?

4) The authors considered only the telecommunication range. However, it is known that hollow-core fibers can operate well beyond the transparency range of quartz glass (even in the mid-IR). So, such fibers may be interest in significantly wider spectral range. I suggest to comment this point (and may be add some additional calculating results).

5) Line 339, “If the refractive index of the quartz is expressed by Snell’s equation…” Refractive index dependence on frequency is described by the Sellmeier equation.

6) Figure 4a. CL are very low for g<15. So, it may be useful to add the same graphs in the logarithmic scale for CL.

7) There are misprints, for example
Figure 4, right (4th) column, the upper and the lower panels labeled as LP_01^4,x. Should the lower panel be labeled as LP_01^4,y?
Line 135, “we used…”, “we” should be changed for “We”.

Reviewer 2 Report

I have following notes to yours manuscript:

1.      The read text of paper is very hard because the authors put into text lot of information about varying parameters (Dij, g, l, k). I suggest put this information to the table with used of interval of its value. It will be clearer for the readers.

2.      Haw the changed results of analyses with small changes of parameters? It is important for real technological preparation of suggest fiber.

3.      Haw were calculated refract indexes of suggest fiber? It is important for field distribution.

4.      In line 135 and 140 are lowercase letters at the beginning of a sentence. Nead to change to capital.

I can recommend the paper for publication if my comments are implemented.

Reviewer 3 Report

The author proposed a novel NMH-ARF with its structural design based on the traditional single-core nested anti-resonant fiber. They calculated the CL, D, and DGD of NMH-ARF under different parameter settings and illustrated the bending resistance and mode extinction characteristics of the fiber. The maximum DGD of different modes reduced to 0.085 ps/km and the maximum CL reduced to 0.008 dB/km under optimized structural parameters. This fiber structure has low sensitivity to bending, and achieved a super-low DGD and a super-low confinement loss (CL) at a wavelength of 1.55 µ m, when keeping CD relatively low. The article is novel in structure, rich in research, and excellent in performance, and I agree to its publication after addressing the following issues

1、 There are too many abbreviated terms in the article, and some of the phrases could be combined and streamlined better. The author should have paid more attention to the readability of the article.

2、 I think the structure schematic can represent the material of each part more clearly. If A, B, C, O in the structure schematic in figure.1 are the multiple hollow cores mentioned by the author, then the white part of the diagram will represent the vacuum and I am curious to know if the author has considered the process feasibility of the structure?

3、 How does the structure shown in Figure. 1(b) constrain the guided modes in the four cores A, B, C, O? Please elaborate on this? How does this structure reflect the anti-resonance?

4、 Usually there is a certain angle of incidence when an optical signal travels long distances in an optical fiber, or as in the case of the bending resistance test mentioned by the author, there is bound to be leakage of modes, and it would be more convincing to give a map of the electric field distribution at this point. I think the bending resistance test mentioned by the author is interesting and I hope that the test can be elaborated more clearly.

5、 About Optical Fiber Sensing, some relevant literature authors need to mention, such as: Phys. Chem. Chem. Phys., 24, 21233 (2022); About “full-vector finite element method”, some relevant literature authors need to mention, such as: Plasmonics 2018, 13, 345–352; Sensors 2022, 22, 6483; Plasmonics 2015, 10, 1537–1543.

6、 There are several references to confinement loss in the article, please elaborate on the effect of fiber length on the transmitted signal?

Round 2

Reviewer 1 Report

The authors responded to my comments and improved the article according to them. So, I recommend the revised version for publication in Photonics.  

Reviewer 3 Report

Accept in present form.